# OpenReview forum: "RULE: Reinforcement UnLEarning Achieves Forget-retain Pareto Optimality"
_NeurIPS.cc/2025/Conference — NeurIPS 2025 poster_

### Official Review · Reviewer_AitF · 2025-06-30

**Clarity:** 3
**Significance:** 3
**Originality:** 2
**Rating:** 5
**Confidence:** 4

**Summary:**

This paper introduces a new LLM unlearning method: RULE, which applies reinforcement learning such as GRPO in LLM unlearning training. By optimizing against a reward considering the unlearning effectiveness based on final text response, this method effectively removes unwanted knowledge and maintain LLM itself.
The experiments show that it outperforms previous methods in improving unlearning performance and maintaining answer naturlness.

**Questions:**

* Previous RL literature mentions length growing in math, what happens to the unlearning training? Does it slow down the generation for final LLM?
* Some previous works also involve synthesizing additional retain data [1,2], what's the difference between the data synthesized in this work?

[1] Who’s Harry Potter? Approximate Unlearning in LLMs https://arxiv.org/pdf/2310.02238

[2] Reversing the Forget-Retain Objectives: An Efficient LLM Unlearning Framework from Logit Difference https://proceedings.neurips.cc/paper_files/paper/2024/hash/171291d8fed723c6dfc76330aa827ff8-Abstract-Conference.html

**Ethical Concerns:**

["NO or VERY MINOR ethics concerns only"]

**Final Justification:**

My concerns are addressed.

**Limitations:**

Please see the weakness.

**Paper Formatting Concerns:**

No.

**Quality:**

2

**Strengths And Weaknesses:**

> Strengths:
* Applying GRPO-like RL in LLM unlearning is reasonable to me.
* The experiment performance is strong. Compared to previous unlearning methods, RULE achieves high language naturalness and unlearning performance.

> Weakness:
* RULE needs a boundary dataset synthesis, which currently is implemented by prompting GPT-4o to replace name entities in the original question/answer pairs. I'm concerned that this procedure introduces noises on other name entities that should be retained in the final model. This phenomenon is also highlighted in a previous work [1].
* RULE heavily relies on the reward design since it is involves RL training. The current reward implementation is mainly a regex string match function as shown in Appendix C. I'm concerned that there may be some form of reward hacking phenomenon, which is very common in previous RL literature.
* I'm not fully convinced by the claim about efficienc improvement. Since RULE is a RL training method, which involves large-scale online response generation process, and takes long time in training. For example, from my personal experience, it takes around 5 min for one step of RL training on 8 H-100 for a 7B LLM. Currently the efficiency is compard through training data size, which is not fair in claiming efficiency from my point of view.

* Minor:
	* Appendix seems not using the Neurips template.
	* Some of the figures are hard to read, e.g., Figure 2 (a) the red performance difference text is hard to differentiate agains background bar, and Figure 3 x/y axis texts are too small.

[1] Revisiting Who’s Harry Potter: Towards Targeted Unlearning from a Causal Intervention Perspective https://arxiv.org/pdf/2407.16997

---

> ### Author Rebuttal · Authors · 2025-07-31
>
> **Feedback**
>
> Thank you for your positive feedback on RL for unlearning and the strong experimental performance; below is our response to the concerns raised.
>
> ---
> **Rebuttal**
>
> > W1. GPT-4o for boundary data construction may introduce noises on other name entities that should be retained in the final model.
>
> Thank you for raising the concern. We acknowledge the potential of noise when constructing boundary data. Here are our thoughts:
>
> - **Why does WHP[1] introduce bias?** As mentioned in [2], WHP[1] introduces hard entity links from the retain set. In contrast, RULE does not introduce such rigid links. We ensure that only semantically similar replacements are made, and we perform manual verification to ensure that the synthesized data remains factual and does not distort the model’s understanding of retained entities.
> - **From an RL perspective:** We do not enforce the model to explicitly learn synthesized knowledge. Rather, RL encourages the model to generate normal response behavior on the synthesized boundary set, which is sampled from the model’s original output distribution. This approach avoids the issues of hard, supervised labels, minimizing the potential negative impact on the retain side, and thereby reducing the risk of introducing noise or unwanted bias in the model’s retained knowledge.
>
> We also experimented with different data synthesis methods with:
>
> - Random selection from the retain set.
> - Semantic similarity selection (`all-MiniLM-L6-v2`) to retrieve themost similar samples from the retain set.
> - Different LLMs following the guideline in our paper.
>
> |Model|Forget|↓|||Retain|↑||
> |-|-|-|-|-|-|-|-|
> |-|Level 1|Level 2|Level 3|Avg.|Level 1|Level 2|Avg.|
> |GPT-4o|30.7|15.3|36.0|27.4|75.7|72.1|73.9|
> |Random selection|37.2|21.0|42.6|33.6|77.9|67.8|72.9|
> |Semantic similarity|9.2|27.4|38.3|25.0|61.7|42.5|52.1|
> |Claude-3.5-sonnet|21.4|13.9|29.0|21.4|67.9|66.1|67.0|
> |Qwen-2.5-7b|34.9|32.6|43.5|37.0|67.3|44.9|56.1|
> |SimNPO|42.1|36.1|42.2|40.1|82.8|70.3|76.5|
>
> Our experiments shows that: In general, the more high-quality the synthesized data, the stronger the model's ability to distinguish, making it easier for the model to generalize. When using smaller models (Qwen-2.5-7b), we found that it often synthesized unrealistic and low-quality data. In this case, the bias you mentioned may indeed occur. We will discuss this in the limitations section.
>
> [1] Revisiting Who’s Harry Potter: Towards Targeted Unlearning from a Causal Intervention Perspective
>
> [2] Who’s Harry Potter? Approximate Unlearning in LLMs.
>
> > W2. RULE heavily relies on reward design. Regex string match function may result in reward hacking phenomenon.
>
> Thank you for raising this concern on reward models.
>
> In terms of the **reward design:**
>
> We made additional experiments to explore the reward variants:
>
> |Method|Forget|↓|||Retain|↑||
> |-|-|-|-|-|-|-|-|
> |-|Level 1|Level 2|Level 3|Avg.|Level 1|Level 2|Avg.|
> |**Intrinsic**||||||||
> |Semantic Similarity|28.3|15.0|36.5|26.6|78.3|65.2|71.7|
> |ROUGE-L|30.7|15.3|36.0|27.4|75.7|72.1|73.9|
> |**LLMs-as-a-judge**||||||||
> |GPT-4o-Mini|26.9|14.8|30.6|24.1|78.8|60.9|69.9|
> |Qwen-2.5-7B|4.9|8.1|17.5|10.2|28.7|19.7|24.2|
> |**Prev. SOTA**||||||||
> |SimNPO|42.1|36.1|42.2|40.1|82.8|70.3|76.5|
>
> Our ROUGE reward offers the best trade-off (Forget: 27.4, Retain: 73.9), while MiniLM similarity is a strong LLM-free alternative (Forget: 26.6, Retain: 71.7). GPT-4o-Mini provides the lowest forget score (24.1) with high retention, while Qwen-2.5-7B over-forgets and retains poorly. These results demonstrate that RULE is robust to reward choices, with simple, non-parametric rewards ensuring scalability and practicality.
>
> As for the **reward hacking**:
>
> We acknowledge the potential risk of reward hacking and appreciate your suggestion. Here are our observations during RULE training:
>
> - **Length-based Reward Hacking:** We observed that the model sometimes tries to generate longer responses to increase the probability of "hitting" the refusal pattern, but this behavior leads to the model recognizing the importance of providing justifications for refusals. This results in safer refusal behaviors, mitigating risks.
> - **Over-Refusal:** A potential reward hacking strategy involves the model rejecting all related queries, but this strategy is curbed by the refusal boundary and the guidance from boundary queries (D̃r). The model is discouraged from indiscriminately refusing and is encouraged to reject only the target queries, preserving utility.
> - **Risk of Hallucination:** One side effect of over-emphasizing retention is that the model might over-remember, leading to hallucination (i.e., generating fabricated information) to get the retain reward. However, this is mitigated by the careful design of boundary data and the reward function, i.e., only above the ROUGE-L threshold, can the model get correct reward, which ensures that the model learns to reject irrelevant knowledge without fabricating answers.
>
> While we acknowledge the potential risk of reward hacking, our current experiments suggest that RULE’s design, particularly boundary queries and non-parametric reward signals, alleviates this issue. We will leave a deeper investigation into reward hacking and its prevention strategies to future work. We hope this detailed analysis addresses your concern regarding reward hacking in the RULE framework.
> > W3. Currently the efficiency is compared through training data size, which is not fair in claiming efficiency from my point of view.
>
> We thank the reviewer for requesting a clearer compute comparison and include a table below.
>
> |Method|Epochs|Tokens|FLOPs|Relative Cost|
> |-|-|-|-|-|
> |RS|2|271,906|6.87P|1.00x|
> |RS+RL|2+1(×8 rollouts)|3,563,744|51.61P|7.52x|
> |GA|3|12,633,024|370.74P|54.00x|
> |NPO|3|12,633,024|370.74P|54.00x|
> |SimNPO|3|12,633,024|370.74P|54.00x|
>
>
> **Why RULE is far cheaper?**
>
> - **Training data size.** GA, NPO, and SimNPO are trained on the original passage corpus along with retain set, hence their token counts are ≈ 14 × larger.
> - **Targeted supervision.** RS uses a *small* set of refusal queries; RL roll-outs (8×) still sample orders-of-magnitude fewer tokens than full-corpus methods.
>
> Note: some of the methods that require reference model are not taken into account.
>
> > Typo. Appendix & Figure (2) text visibility, Figure (3) x/y axis size.
>
> Thanks for pointing out these Typos, we will revise the mentioned Typo in the final version.
>
> > Q1. What happens to the unlearning training? Does it slow down the generation for final LLM?
>
> Thank you for raising this insightful question. We performed an analysis on the average response length, and the results were quite interesting:
>
> |Phase|Avg. Forget|% Change|Avg. Retain|% Change|
> |-|-|-|-|-|
> |Before|44.81||28.51||
> |After RS.|33.77|-24.7%|26.15|-8.3%|
> |After ReBO.|99.60|122.27%|82.60|189.72%|
>
> **Why does this happen?** Our manual inspection of the model's output revealed the following insights:
>
> - **After RS**: The model transitions from providing complete answers to issuing shorter refusals during the Rejection Steering (RS) phase, reducing response length. (e.g., "I don't have information on Stephen King").
> - **After ReBO**: In the ReBO phase, response length increases as the model provides more detailed justifications for refusal and explanations for retain set.
>
> While RL may increase response length and inference time, the model's self-aware justification of refusals enhance safety. We will mention the potential cost associated with this increase in the limitations section. Thank you for your feedback!
>
> > Q2. Some previous works also involve synthesizing additional retain data, what's the difference between the data synthesized in this work?
>
> We appreciate the reviewer’s concern regarding the data synthesis strategy used in this work. While some previous works, such as [1] and [2], also synthesize additional retain data to aid the unlearning process, our approach differs significantly in how we handle data synthesis.
>
> Specifically, [1] synthesizes additional "anchor terms" by replacing unique terms in the data with more generic ones (e.g., replacing "Harry" with "Jon"). It builds misleading knowledge links that guide the model towards unlearning specific content. While effective, it may introduce a risk of hallucination by generating knowledge that could mislead the model into over-generalizing or losing critical factual context.
>
> In [2], the authors introduce a framework for unlearning through logit differences, which also involves generating additional perturbed retain data. This synthesized data is used to train an “assistant model” that incorporates misleading knowledge for the purpose of calculating logit differences. Like [1], this approach relies on misleading knowledge, which could potentially result in the model retaining incorrect or fabricated information during the unlearning process.
>
> In contrast, our approach avoids the generation of misleading data. Instead, we focus on synthesizing boundary queries (D̃r) that are both challenging and factually accurate. These queries are designed to explore the refusal boundary, guiding the model to properly reject forget-related content while still generating useful and factual information for the retained knowledge. This approach ensures that the model's outputs remain grounded in factual knowledge, reducing the risk of hallucination.
>
> We hope this response adequately addresses the reviewer’s concern regarding the differences in data synthesis strategies.
>
> [1] Who’s Harry Potter? Approximate Unlearning in LLMs
>
> [2] Reversing the Forget-Retain Objectives: An Efficient LLM Unlearning Framework from Logit Difference
>
> ---
> **Summary**
>
> We appreciate the reviewer’s feedback and recognition of RULE. We have discussed the concerns regarding boundary data, reward design, and computational efficiency. We hope that our responses have addressed the reviewer’s concerns adequately. Please let us know if any additional clarifications are needed.

---

> > ### Comment · Reviewer_AitF · 2025-07-31
> >
> > Thanks for the detailed response. I'm glad to see the additional experiments, and they addressed my concerns. I have increased the rating and suggest including these discussions in the final version.

---

> > > ### Author Response · Authors · 2025-08-01
> > >
> > > We sincerely appreciate the reviewer’s thoughtful feedback and constructive suggestions. We are grateful for the time and effort spent reviewing our work. Your insightful comments have significantly contributed to improving the quality of our paper.

---

### Official Review · Reviewer_esTn · 2025-06-30

**Clarity:** 3
**Significance:** 2
**Originality:** 3
**Rating:** 4
**Confidence:** 3

**Summary:**

The paper introduces RULE, a new method for unlearning in large language models (LLMs). RULE leverages a sample-efficient, reinforcement learning (RL)-based training pipeline designed to enforce high-quality refusal responses for inputs similar to the forget set, while maintaining appropriate replies for content intended to be retained.

The training pipeline consists of several steps:
1. Teaching Refusal Behavior: Since refusal behavior is uncommon in most pre-trained models, the first step is to train the model to effectively refuse certain requests.
2. Constructing a High-Quality Refusal Boundary: This involves generating a boundary set using a SOTA model (GPT-4o). The boundary set contains samples that should be retained but closely resemble the forget set, helping to delineate when to refuse.
3. Reinforcement Learning Training: The model undergoes on-policy RL training. The reward mechanism encourages refusal of responses related to the forget set, including relevant disclaimers (e.g., “I cannot provide an answer relating to this copyrighted work of Stephen King”). For the boundary set, the model aims to preserve responses close to those of the original model, with similarity assessed using the ROUGE-L metric.

The authors' findings highlight that RULE delivers superior refusal quality for the forget set compared to baseline methods, even with substantially less data. While this comes at the expense of reduced retention quality, empirical results suggest that RULE strikes an efficient balance between forgetting and retaining content. Despite these significant benefits, the pipeline suffers from general complexity, with the construction of the boundary set serving as a crucial insufficiently explored step.

**Questions:**

1. Have you considered how the effectiveness of RULE might vary depending on the LLM used to construct the boundary set? For example, what differences emerge when using:
- Reasoning models from online vendors
- Local models
- Models specifically fine-tuned to create high-quality boundary sets
2. Additionally, how does the cost-benefit trade-off of RULE shift when factoring in the inference costs associated with the LLM used for constructing the boundary set?
3. Regarding refusal steering, what would be the optimal size for the forget set, and how many iterations does this process typically require compared to the main RL loop? Does this vary with the model size?
4. When considering the overall pipeline, how does it scale relative to the sizes of both the forget set and the model itself?
5. During experimentation, did you explore simpler methods for various components?
- For instance, instead of synthetically generating the boundary set, you could use sentence embeddings to identify similar pairs between the forget and retain sets, using that retain subset as the boundary set.
- Alternatively, leveraging the model’s hidden state to assess similarity to the original model—or to specific entities for the forget set—could offer valuable insights. For example, using "Washington" as a metonym may not be effectively captured by the current approach, but could be when using embeddings.

**Ethical Concerns:**

["NO or VERY MINOR ethics concerns only"]

**Final Justification:**

The authors systematically ablated various components of the intricate RULE pipeline, demonstrating the effectiveness of their choices and presenting appealing alternatives for practitioners (e.g., switching from black-box LLMs if preferred). Additionally, they provided information on the computational cost of their method and its resilience to various types of attacks, amongst other, more minor requests.

In conclusion, the rebuttal process has significantly improved the submission, prompting me to raise my score. My central reservations remain: a) the complexity of the pipeline, which may hinder adoption, and b) the inherent limitation of refusal-based unlearning (the inability to guarantee that the data is forgotten). Given the author’s comprehensive rebuttal and the significant number of ablation experiments included in the camera-ready version, I recommend its acceptance by the AC. I believe this work will contribute to a broader understanding of the field, especially for designing RL-based unlearning methods.

**Limitations:**

yes

**Quality:**

3

**Strengths And Weaknesses:**

Strengths
1. Principled Decision Boundary: Enforcing a decision boundary effectively enhances the forget-retain trade-off.
2. Meaningful Refusal Answers: The approach of providing semantically meaningful refusal answers is superior to uncontrolled responses.
3. Theoretical Foundation: The method’s effectiveness is theoretically supported, as detailed in the appendix.
4. High-Quality Writing: The manuscript is well-written and comprehensive in form.
5. Reproducibility: The availability of code and a detailed appendix facilitates easy reproduction of the work.

Major Weaknesses
1. Dependency on Manual Annotation or LLM: The need for manual annotation or LLM-based generation for the boundary set introduces potential confounding factors, with limited exploration of the LLM-PROMPT pair's impact on RULE performance.
2. Training Data and Optimization Guidance: The paper lacks clear guidance on the quantity of training data and the number of optimization iterations required for the Rejection-Steering phase, which are critical for pipeline effectiveness.
3. The method is generally complex and involves an ample design space for all proposed components, making it potentially brittle at scale.

Minor Weaknesses
1. Opaque Qualitative Evaluation: Utilizing an LLM as a judge for qualitative assessments reduces interpretability.
2. Limited Similarity Metrics: Relying solely on ROUGE-L for similarity assessment without reference to the model's latent space may limit the effectiveness of the reward signal. For example, it is unclear what the reward should be for answering with the same content as the base model, but in a different language.
3. Scalability of Refusal Templates: The linear growth of predefined refusal templates with additional languages necessitates either manual translations or automated translation tools.
4. Lack of Computational Cost Analysis: I could not find an indication of how RULE’s computational cost scales with the forget set size and model size. The appendix provides raw values for the number of GPU hours required for each phase.
5. No Exact Unlearning Baseline: The paper does not compare with exact unlearning, which involves re-training the model with the forget set completely excluded.

Typos:

1. Line 117: “making it difficult to ensure the utility” -> “making it difficult to ensure the utility of the model.”
2. Line 144: “We introduce a set of boundary set” -> “We introduce a boundary set.”

---

> ### Author Rebuttal · Authors · 2025-07-31
>
> **Feedback**
>
> Thank you for recognizing our innovation, theoretical foundation, and reproducibility. Below are responses to the reviewer’s concerns.
>
> ---
> **Rebuttal**
> > Major W1. Dependency on LLM for the boundary set; impact of the LLM-PROMPT pair is under-explored.
>
> We appreciate the reviewer’s insightful concern regarding LLM-generated boundary data. While entity-replacement with a single LLM is an effective method for generating hard negatives, we agree that relying on one LLM may limit generality. We validated variants with identical settings in our paper:
>
> - Random selection from the retain set.
> - Similarity selection (`all-MiniLM-L6-v2`) to retrieve the most similar samples from the retain set.
> - Different LLMs constructions following the guideline in our paper.
>
> |Model|Forget|↓|||Retain|↑||
> |-|-|-|-|-|-|-|-|
> |-|FB|QA|AA|Avg.|FB|QA|Avg.|
> |GPT-4o(default)|30.7|15.3|36.0|27.4|75.7|72.1|73.9|
> |Intrinsic||||||||
> |Rand. Select|37.2|21.0|42.6|33.6|77.9|67.8|72.9|
> |Similarity|9.2|27.4|38.3|25.0|61.7|42.5|52.1|
> |LLMs Construction||||||||
> |Claude|21.4|13.9|29.0|21.4|67.9|66.1|67.0|
> |Qwen-2.5-7b|34.9|32.6|43.5|37.0|67.3|44.9|56.1|
> |Baseline||||||||
> |SimNPO|42.1|36.1|42.2|40.1|82.8|70.3|76.5|
>
> Compared to the GPT-4o baseline, we observe that:
> - **Intrinsic methods** yield relatively competitive results, both achieving comparable forget and retain scores.
> - **LLMs Construction**: Claude-3.5-Sonnet performs well, while Qwen-2.5-7b shows lower performance, indicating that the choice of LLM can significantly impact the quality of boundary data.
>
> These results demonstrate that RULE remains robust regardless of how the hard negatives are obtained.
> > Major W2. No guidance on the amount of training data/iterations for Rejection Steering.
>
> Thank you for highlighting the need for implementation guidance. We will certainly add a dedicated section in `App.D`. The details are listed below:
>
> |Dataset|LR|Batch Size|Epochs|#Data|
> |-|-|-|-|-|
> |RWKU|4e-7|8|2.0|300|
> |MUSE|4e-7|8|1.0|1682|
> > Major W3. Pipeline Complexity and Scalability.
>
> We appreciate the reviewer's concern regarding RULE's complexity and scalability. Our method aims to minimize moving parts while achieving effective unlearning. Below is our response to clarify the pipeline.
>
> **1. Pipeline Design Rationale**
>
> - **Training Phase**
>
> Warm-start is standard before RL: without a short SFT, the policy rarely explores the target behavior when it lies far outside the base model distribution[1]. Our ablation (`response to reviewer BvY9 W1.Table R1`) shows that removing RS slashes refusal reward from 0.83 → 0.02, indicating that RS is a minimal but indispensable bridge, which enables the RL stage to explore and optimize near the refusal boundary.
> - **Boundary Data Construction**
>
> RULE avoids the need for a large, annotated retain set by using entity replacement to synthesize hard negatives, making it simple and scalable with off-the-shelf LLMs. Inspired by recent RL for reasoning works[2], our goal is to teach models with hard cases for better generalization to easier ones. Experiments (reported in `Major W1`) show the importance of hard negatives. Low-cost alternatives, like MiniLM-based retrieval, perform well without LLM calls, and replacing GPT-4o with other models leads to minimal performance changes.
> - **Reward Function**
>
> We use an intrinsic reward to avoid the need for training a reward model. We also validated alternative reward variants, as shown in the table below:
>
> |Method|Forget|↓|||Retain|↑||
> |-|-|-|-|-|-|-|-|
> |-|FB|QA|AA|Avg.|FB|QA|Avg.|
> |Similarity|28.3|15.0|36.5|26.6|78.3|65.2|71.7|
> |ROUGE-L|30.7|15.3|36.0|27.4|75.7|72.1|73.9|
> |GPT-4o-Mini|26.9|14.8|30.6|24.1|78.8|60.9|69.9|
> |Qwen-2.5-7B|4.9|8.1|17.5|10.2|28.7|19.7|24.2|
> |SimNPO|42.1|36.1|42.2|40.1|82.8|70.3|76.5|
>
> The default ROUGE-based reward offers the best trade-off, while `all-MiniLM-L6-v2` similarity is a strong LLM-free alternative. GPT-4o-Mini provides the lowest forget score with high retention, while Qwen-2.5-7B over-forgets and retains poorly.
>
> These results demonstrate that RULE is robust to reward choices, with simple, non-parametric rewards ensuring scalability and practicality.
>
> **2. Scalability of RULE.**
>
> Because of limited computational budget and rebuttal time, we conducted experiments using **LLaMA-3.2-1B** and **LLaMA-3.2-3B**:
>
> |Model|Forget|↓|||Retain|↑||
> |-|-|-|-|-|-|-|-|
> |-|FB|QA|AA|Avg.|FB|QA|Avg.|
> |**LLaMA-3.2-1B**||||||||
> |Before|35.4|28.4|39.2|34.3|31.7|35.3|33.5|
> |RS|34.4|28.3|35.4|32.7|31.7|35.5|33.6|
> |ReBO|31.1|24.1|31.5|28.9|33.7|35.8|36.1|
> |**LLaMA-3.2-3B**||||||||
> |Before|53.0|51.7|53.3|52.7|60.3|55.0|57.7|
> |RS|51.8|45.9|51.3|49.7|60.4|55.7|58.1|
> |ReBO|47.2|31.0|42.2|40.1|58.2|50.4|54.3|
>
> The results show that smaller models retain the trends from our main experiments, indicating that RULE’s behavior is not reliant on large model capacities. Additionally, we demonstrate in the main paper that RULE transfers effectively across various **RL algorithms** and **model variants**.
>
> [1] Deepseek-r1: Incentivizing reasoning capability in llms via reinforcement learning.
> [2] Behavior Injection: Preparing Language Models for Reinforcement Learning.
> > Minor W1. Opaque Qualitative Evaluation
>
> We share the reviewer’s concern and have made the LLM-judge as transparent as possible. The full naturalness evaluation rubric and GPT-4o prompts (App.D.3) are published, and the LLMs provide rationales for their scores, which we manually verify. We are adding manual annotations for correlation with human judgments, and we will include the full evaluation results before the `Reviewer Author Discussions` period ends. Upon acceptance, we will release all judged outputs and LLM reasoning traces for independent review.
> > Minor W2. Limited Similarity Metrics & Language
>
> We agree that relying solely on ROUGE-L is not reliable. We have explored other reward functions in response to `Major W3`. Since prior unlearning and RL works mostly focus on English, we aimed for consistency.
> > Minor W3. Scalability of Refusal Templates
>
> We implemented TOFU’s 100 refusal templates, converted to regex patterns, with matching accuracy over 95% (`App.C.2`). We focused on English because existing benchmarks are in English. Our additional reward functions in `Major W3` provide language-agnostic options, making template scalability not a bottleneck and supporting multilingual extensions.
> > Minor W4. Lack of Computational Cost Analysis
>
> We thank the reviewer for requesting a clearer computation comparison and include a table below.
>
> |Method|Epochs|Tokens|FLOPs|Relative Cost|
> |-|-|-|-|-|
> |RS|2|271,906|6.87P|1.00x|
> |RS+RL|2+1(×8 rollouts)|3,563,744|51.61P|7.52x|
> |GA|3|12,633,024|370.74P|54.00x|
> |NPO|3|12,633,024|370.74P|54.00x|
> |SimNPO|3|12,633,024|370.74P|54.00x|
>
> Note: some methods that require reference models are not taken into account.
> > Minor W5. No Exact Unlearning Baseline
>
> We thank the reviewer for noting the retrain baseline. There are two practical constraints of the benchmarks we use:
>
> - RWKU: The benchmark intentionally omit a clean retain set and discourage retraining, as it’s often infeasible in real-world scenarios. As such, no RWKU papers report such a baseline, and we followed this to maintain fairness.
> - MUSE: We re-implemented all unlearning methods compared in the original MUSE paper. Due to computational budget, we will include the retrain performance reported in the original paper:
>
> > Q1. Effectiveness of RULE might vary depending on the LLM used to construct the boundary set?
>
> Please refer to our response to `Major W1`.
> > Q2. Cost-benefit trade-off of RULE shift when factoring in the inference costs associated with the LLM used for constructing the boundary set?
>
> To quantify the trade-off, we experimented with three LLMs. Results in `Major W1` show that stronger models like GPT-4o and Claude generate better boundary data, improving unlearning behavior, while Qwen-2.5-7b's noisy annotations degrade performance. GPT-4o provides the best balance of performance and cost, with Claude-3.5-Sonnet as a competitive alternative. Qwen-2.5-7b, while free locally, significantly impacts effectiveness. For 300 queries, GPT-4o and Claude-3.5-Sonnet costs approximately $0.2–0.5, while Qwen is free if run locally.
> > Q3. The optimal size for the forget set & iterations for RS compared to the main RL loop? Does this vary with the model size?
>
> Due to computational constraints, searching optimal size would require significant resources, which is an O(4) complexity (data size × model size × RS iter × RL iter). We used the original training data but can share details on how varying RS epochs affect performance (RL steps are fixed to 20 steps):
>
> |Epochs|Forget ↓||||Retain ↑|||
> |-|-|-|-|-|-|-|-|
> ||FB|QA|AA|Avg.|FB|QA|Avg.|
> |**LLaMA-3.2-1B**||||||||
> |1|28.2|21.7|37.2|29.0|29.2|36.8|33.0|
> |2|31.1|24.1|31.5|28.9|33.7|35.8|34.7|
> |3|32.5|27.2|33.8|31.1|33.0|39.1|36.1|
> |**LLaMA-3.2-3B**||||||||
> |1|49.9|33.6|47.3|43.6|60.3|52.7|56.5|
> |2|47.2|31.0|42.2|40.1|58.2|50.4|54.3|
> |3|50.0|36.4|47.7|44.7|57.7|55.2|56.5|
> |**LLaMA-3.2-8B**||||||||
> |1|35.2|28.5|44.3|36.0|77.9|63.7|70.8|
> |2|28.0|16.8|38.3|27.7|76.2|71.3|73.7|
> |3|31.5|24.3|43.7|33.1|79.1|69.9|74.5|
>
> We found that the number of RS epochs affects model performance, with optimal results achieved at epoch 2. We will include the full training dynamics plots in the revised version.
> > Q4. & Q5.
>
> Please see `Major W3` above.
>
> ---
> **Summary**
>
> We have discussed the boundary data, reward design, and the scalability of RULE with experimental results to support our claims and clarified the scalability of RULE. We will include additional details including the training dynamics, discussion on the rationale of design choices, and computation comparison in the final version.
>
> Due to space constraints, we haven't included all rebuttals in detail. We’re happy to address any concerns and appreciate the opportunity to clarify our work. Please let us know if any additional clarifications are needed.

---

> ### Comment · Reviewer_esTn · 2025-08-01
>
> Thank you very much for this comprehensive rebuttal. The new experiments and arguments you’ve added will significantly strengthen the submission, addressing my major concerns about the methodology of the paper.
>
> While the complexity of the pipeline may pose a significant obstacle to practical adoption, I believe the paper constitutes a valuable scientific contribution, and I’m inclined to raise my score.
>
> I have one final question: given that your method guides the model towards refusing rather than explicitly attempting to remove the forgotten data, do you have an idea of how it would perform against common black/white-box attacks? For instance, could the forgotten information be retrieved through an indirect prompt (“Who is the author of that book about vampires in Maine”) or by directly examining the weights/activations?

---

> > ### Author Response · Authors · 2025-08-04
> >
> > We sincerely appreciate the reviewer’s thoughtful feedback and constructive suggestions, which have contributed to the improvement of our paper's quality.
> >
> > Here is a response to your question about **adversarial unlearning** of RULE.
> >
> > - **Blackbox attacks**: RWKU provides adversarial attack prompts built upon traditional QA that contain misleading queries to test if the knowledge will be elicited by adversarial prompt attacks, including prefix injection, affirmative suffix, role playing, reverse query, and others.
> >
> > - **Whitebox attacks**: We reported "relearning attacks" which re-finetune the forget set to the unlearned model. And we also re-implemented the "Enhanced GCG" in https://github.com/ethz-spylab/unlearning-vs-safety.
> >
> > |Attack Type       | Before |   RS   |  ReBO  |
> > |----------------------------|--------|--------|--------|
> > | **Baseline**               |        |        |        |
> > | Forget QA ↓                |  70.3  |  43.0  | **16.8** |
> > | **Black-box**              |        |        |        |
> > | RWKU Adv.QA ↓              |   –    | 51.2 (+8.2)  | 38.3 (+21.5) |
> > | **White-box**              |        |        |        |
> > | Retrain + Forget QA ↓      |   –    | 52.4 (+9.4) | 26.8 (+10.0)|
> > | GCG Adv.QA ↓               |   –    | 62.1 (+19.1) | 46.7 (+29.9) |
> >
> > Takeaways:
> >
> > - Adversarial attacks are partially effective: Our RL-based pipeline generalises beyond the seen forget queries and reduces leakage under both black-box attacks, but it does not eliminate it entirely. Even so, the forget performance we report is markedly lower than prior SOTA.
> >
> > - Re-training cannot undo the refusal behaviour: Because RULE optimises for refusal rather than hard knowledge deletion, re-introducing the forget set during fine-tuning fails to make the model answer: the learned refusal policy is largely preserved.
> >
> > - Enhanced GCG still breaks the defence: The gradient-guided prefix attack can recover a non-trivial portion of the forgotten information, highlighting the need for dedicated counter-measures against prefix-based jailbreaks in future work.
> >
> > We hope these results fully address the reviewer’s concerns about RULE’s resilience to common black- and white-box adversarial attacks.

---

> > > ### Comment · Reviewer_esTn · 2025-08-04
> > >
> > > Thank you very much, I have no further questions.

---

> ### Author Response · Authors · 2025-08-06
>
> We are grateful for the time and effort spent reviewing our work. Your insightful comments have significantly contributed to improving the quality of our paper.
>
> We've made an human judgement to naturalness with 100 randomly sampled model outputs:
>
> | Metrics      | Pearson *r* | Spearman ρ | Kendall τ |
> |----------------|------------:|-----------:|----------:|
> | Readability    | **0.702**   | **0.690**  | **0.485** |
> | Helpfulness    | **0.502**   | **0.492**  | **0.341** |
> | Truthfulness   | **0.891**   | **0.721**  | **0.714** |
>
> - Truthfulness shows very strong agreement, as evaluators rarely dispute basic facts.
>
> - Readability also enjoys solid consensus, reflecting shared intuition about fluency and grammar.
>
> - Helpfulness lags because it must juggle user intent, completeness, policy compliance, and tone, factors that different judges weigh differently.

---

### Official Review · Reviewer_BvY9 · 2025-07-02

**Clarity:** 3
**Significance:** 3
**Originality:** 2
**Rating:** 4
**Confidence:** 4

**Summary:**

This paper addresses the critical problem of targeted unlearning in Large Language Models (LLMs), where specific information must be removed from a model without compromising its overall utility. The authors propose a novel framework, Reinforcement UnLEarning (RULE), which formulates unlearning as a boundary optimization task solved via reinforcement learning. RULE leverages a small forget set and synthesized boundary queries, guided by a reward function that balances safe refusal and helpfulness.

**Questions:**

See weakness.

**Ethical Concerns:**

["NO or VERY MINOR ethics concerns only"]

**Final Justification:**

It is the first attempt to formulate LLM unlearning as refusal boundary optimization via reinforcement learning. It has strong emprical results.

**Limitations:**

Yes

**Quality:**

3

**Strengths And Weaknesses:**

Strength:

1.	The authors have innovatively explored the application of reinforcement learning in the context of unlearning.
2.	The authors’ method shows a notable improvement compared to the baseline.
3.	The paper provides both theoretical and empirical evidence that RULE achieves a superior trade-off between forgetting target information and preserving general model utility.
4.	The paper demonstrates remarkable data efficiency, using "only 12% forget set and 8% synthesized boundary data" compared to baselines that require 100% of both forget and retain datasets. This makes it more practical for real-world scenarios where large, cleanly partitioned datasets are often unavailable.

Weakness:

1. The reward design proposed by the authors appears to suffer from very sparse rewards. Have the authors attempted to address this issue?
2. In the Step 3 of Appendix B, The conditions for applying the martingale inequality do not appear to have been proven to hold.

---

> ### Author Rebuttal · Authors · 2025-07-31
>
> **Feedback**
>
> Thank you for your time and thoughtful review of our paper. We are encouraged to see that our **exploration on RL for unlearning, data efficiency, and potential real-world application** has been recognized. Below, we provide a detailed rebuttal addressing the reviewer’s concerns.
>
> ---
> **Rebuttal**
> > W1. The reward design proposed by the authors appears to suffer from very sparse rewards. Have the authors attempted to address this issue?
>
> We thank the reviewer for pointing out the potential impact of sparse rewards. Sparse rewards are indeed a pervasive obstacle whenever the downstream behavior (in our paper, ***safe refusal***) lies far outside the base model’s training distribution during RL [1,2]. A simple and effective solution is to apply supervised fine-tuning (SFT, warm start) for such distribution shifts [3] before RL, which is also the motivation for the **rejection-steering** phase. We ablated cold start in our ablation studies (`§4.3, Table 2`), which actually aligns well with previous studies.
>
> We also show the average rewards in the validation set that models gain for better illustration:
>
> **Table R1: Ablations for Reward Sparsity.**
> | Variant | **Performance** |  |  | **Rewards** |  |  |
> | --- | --- | --- | --- | --- | --- | --- |
> |  | Forget ↓ | Natural ↑ | Retain ↑ | avg.refusal | avg.retain | avg.total |
> | **Original** | 76.9 | 70.6 | 87.6 | - | - | - |
> | **RULE_GRPO** | 27.7 | 89.1 | 73.7 | 0.8298 | 0.9750 | 0.9112 |
> | **w/o RS** | 71.4 | 65.7 | 85.2 | 0.0213 | 1.0 | 0.6075 |
> | **w/o RS*** | 44.2 | 66.9 | 65.5 | 0.2766 | 0.8667 | 0.5701 |
> | **w/o boundary** | 19.9 | 25.4 | 23.6 | 0.7979 | 0.1500 | 0.4345 |
>
> And the conclusions are:
>
> - **Rejection-Steering (RS) Warm-up mitigates refusal reward sparsity**: Without RS, the model faces refusal reward sparsity, as shown by the **w/o RS** variant (Forget ↓ 71.4, Natural ↑ 65.7, Retain ↑ 85.2) and low reward values (avg. forget_reward = 0.0213).
> - **Cold start with system prompt slightly alleviates refusal reward sparsity**: While the system prompt helps guide positive sampling, it still leads to suboptimal results, as seen in the **w/o RS**variant (Forget ↓ 44.2, Natural ↑ 66.9, Retain ↑ 65.5) with avg. forget_reward = 0.2766, compared to full RS warm-up.
> - **Without boundary data, the model faces retain reward sparsity**, which results in poor performance, as demonstrated by the **w/o boundary** variant (Forget ↓ 19.9, Natural ↑ 25.4, Retain ↑ 23.6) and low avg. retain_reward = 0.1500.
>
> That’s why the **rejection steering** phase is crucial for addressing the **sparse rewards** issue:
>
> - Although the model does not initially recognize the unlearning target, even a small amount of rejection steering ($12%$ forget set) helps the model identify the privacy issue at hand.
> - Through its own exploration, guided by the appropriate rewards, the model learns to generate the correct refusal behavior.
>
> This mechanism enables the model to navigate the distribution shift from normal behavior to refusal, ensuring efficient learning with minimal supervision.
>
> We hope this explanation, along with the experimental results, adequately addresses the reviewer’s concern regarding sparse rewards!
>
> [1] Guo, Daya, et al. "Deepseek-r1: Incentivizing reasoning capability in llms via reinforcement learning." *arXiv preprint arXiv:2501.12948* (2025).
>
> [2] Wang, Zengzhi, et al. "Octothinker: Mid-training incentivizes reinforcement learning scaling." *arXiv preprint arXiv:2506.20512* (2025).
>
> [3] Cen, Zhepeng, et al. "Behavior Injection: Preparing Language Models for Reinforcement Learning." arXiv preprint arXiv:2505.18917 (2025).
>
>
> > W2. In the Step 3 of Appendix B, the conditions for applying the martingale inequality do not appear to have been proven to hold.
> >
>
> Thank you for pointing out this omission. The martingale-concentration step is valid in our setting; below we supply the missing assumptions and a concise proof sketch.
>
> **Theorem.**
>
> Let the retain-side risk be estimated via an on-policy reinforcement learning process collecting data from $K$ updates, with $m$ boundary prompts per update, each generating a rollout of length $H$. The estimation error, $\epsilon_{\text{EXPLORE}}$, between the true retain-side risk $\Delta_r$ and the empirical risk $\hat{\Delta}_r$, is bounded with probability at least $1-\delta$ as:
>
> $$\epsilon_{\text{EXPLORE}} = |\hat{\Delta}_r - \Delta_r| = \mathcal{O}\left(\sqrt{\frac{\log(1/\delta)}{K m H}}\right)$$
>
> **Proof.**
>
> Let  $N = K\,m\,H$ be the total number of token–level observations collected
> during exploration.  For each token define  $Z_t\in\{0,1\}$ where $Z_t = 1$ iff the policy produces a **false
> refusal** on that token, and let  $\Delta_r := \mathbb{E}\,[Z_t]$ be the true retain-side risk.
>
> The empirical estimate is  $\hat{\Delta}_r := \frac{1}{N} \sum _ {t=1} ^ {N} Z_t.$
>
> Because the roll-out log induces a natural filtration $(\mathcal F_t)$ and $Z_t$ is $\mathcal F_t$-measurable, the Doob martingale
>
> $X_t := \mathbb{E}\left[\sum_{i=1}^{N} Z_i \,\big|\,\mathcal F_t\right]$ satisfies bounded increments  $|X_t-X_{t-1}| \le 1$ (since $Z_t\in[0,1]$).
>
> Applying the Azuma–Hoeffding inequality yields
> $
> \Pr\left(
> \left|N(\hat{\Delta}_r-\Delta_r)\right|
> \ge \sqrt{2N\log\frac{2}{\delta}}
> \right)
> \;\le\;\delta .
> $
>
> Dividing by $N$ and substituting $N=K\,m\,H$ gives, with probability $1-\delta$,
>
> $$
> \boxed{
> |\hat{\Delta}_r-\Delta_r|
> \;\le\;
> \sqrt{\frac{\log(2/\delta)}{2\,K\,m\,H}}
> }
> $$
>
> Hence
>
> $$
> \epsilon_{\text{EXPLORE}}(K,m,H,\delta)
> = \mathcal{O}\left(
> \sqrt{\frac{\log(1/\delta)}{K\,m\,H}}
> \right).
> $$
>
> These additions will make the martingale step fully rigorous, and we will include a full lemma in the final version. We appreciate the reviewer’s attention to this detail and welcome further questions!
>
> ---
> **Summary**
>
> We appreciate the reviewer’s positive feedback on our work and the recognition of RULE's potential. We have addressed the concerns regarding reward sparsity and martingale-concentration, providing a detailed explanation and supporting evidence. We will include the missing assumptions and proof sketch in the final version of the paper. We are open to any further discussion and hope that our responses have addressed the reviewer’s concerns adequately. Please let us know if any additional clarifications are needed.

---

> > ### Comment · Reviewer_BvY9 · 2025-08-08
> >
> > Thank you for your response, which partly addressed my concerns. I will keep my score.

---

### Note · Authors · 2025-08-11

We thank all reviewers for their thoughtful feedback, which has greatly helped us improve the paper.

**Strengths recognized:**

- **Novelty**: First to formulate LLM unlearning as refusal boundary optimization via reinforcement learning (`BvY9, AitF`) with carefully designed boundary data and refusal rewards (`esTn`).

- **Data Efficiency**:  Even with 12% synthesized forget data and 8% boundary data, **RULE** enables the model to explore the refusal boundary and generalize to unseen distribution, which achieves a better forget-retrain trade-off (`BvY9, esTn`).

- **Strong empirical results**: Higher unlearning quality and naturalness than prior methods (`BvY9, AitF`).

- **Theoretical foundation**: Supported by formal analysis with proofs and assumptions stated in the appendix (`BvY9, esTn`).


**Main concerns & clarifications:**

- **Ablations on reward designs** (`BvY9, esTn, AitF`): We discussed and ablated alternatives (e.g., similarity-based, LLMs-as-a-Judgem, etc.) to prove the effectiveness of RULE when using different reward designs.

- **Ablations on boundary set synthesize** (`esTn, AitF`): Added experiments with different LLMs and similarity-retrieval approaches, confirming RULE’s robustness to boundary set source and prompt choice.

- **Computational efficiency & Scalability claims** (`esTn, AitF`): Clarified our computational efficiency compared to existing methods and discussed scaling with model and forget set size.

We believe these additions address the key concerns and further strengthen the paper’s technical contribution and practical value.

---

### Decision · Program_Chairs · 2025-09-17

**Decision:**

Accept (poster)

**Comment:**

Summary:

This paper introduces Reinforcement UnLEarning (RULE), a novel framework for Large Language Model (LLM) unlearning. The central claim is that unlearning can be effectively framed as a refusal boundary optimization problem, solved via reinforcement learning (RL). The proposed two-stage method first performs "Rejection Steering" (a supervised warm-start) to teach the model basic refusal behavior, then uses on-policy RL for "Refusal Boundary Optimization" (ReBO) on a small set of forget queries and a synthetically generated boundary dataset. The key findings, supported by experiments on the RWKU and MUSE benchmarks, are that RULE achieves a superior trade-off between forgetting unwanted knowledge and retaining general utility (i.e., forget-retain Pareto Optimality). Furthermore, the authors claim that RULE is highly data- and compute-efficient, significantly improves the "naturalness" of model responses to forgotten queries, and generalizes refusal behavior to unseen but semantically related topics.

Strengths:

1. Novel and Innovative Formulation: The paper's primary strength is its innovative application of reinforcement learning to the unlearning problem, reframing it as a task of learning a controlled, meaningful refusal behavior.

2. Strong Empirical Performance and Forget-Retain Trade-off: The method demonstrates strong empirical results, consistently outperforming baselines in achieving a superior balance between forgetting target information and retaining general utility, particularly in producing natural responses.

3. Remarkable Data Efficiency: The method is highly data-efficient, requiring only a small fraction of the forget and retain data compared to other methods, which increases its practical applicability.

4. Theoretical Support and Paper Quality: The work is supported by a theoretical foundation and was praised by reviewers for its high-quality writing and the inclusion of details needed for reproducibility.

Weaknesses:

1. Pipeline Complexity and Lack of Guidance: The overall pipeline was noted for its complexity, which could make it difficult to implement and potentially "brittle at scale." The paper also lacked clear guidance on hyperparameters for certain stages.

2. Inherent Limitation of the Refusal-Based Approach: The refusal-based approach primarily teaches the model a behavior (to refuse answering specific queries) but does not guarantee that the corresponding information is truly erased from the model's weights.

Summary of Rebuttal and Changes

Concern 1: Robustness of the Core Methodology

Points Raised: Reviewer AitF and Reviewer esTn questioned the reliance on a single proprietary LLM (GPT-4o) for synthesizing boundary data, fearing it could be noisy or not generalizable. Additionally, Reviewer BvY9 raised the issue of potential sparse rewards from the reward function, while Reviewer AitF was concerned about reward hacking.

Authors' Response: To address these points, the authors conducted a comprehensive set of new ablation studies. They tested RULE's performance using different boundary data sources, including other LLMs (Claude, Qwen-2.5-7b) and non-LLM methods (random selection, semantic similarity), proving the framework was not dependent on a single model. They also ran experiments with alternative reward functions and demonstrated that the "Rejection-Steering" phase effectively mitigates reward sparsity by acting as a warm-start.

Final Decision Weight: This was the most critical part of the rebuttal. By demonstrating that the core methodology was robust and not dependent on a specific proprietary model or a fragile reward design, the authors successfully addressed the primary technical concerns of Reviewers esTn and AitF. This evidence was a major factor in the final decision.

Concern 2: Computational Efficiency Claims

Points Raised: Reviewer AitF argued that the paper's claim of computational efficiency was not well-supported, as comparing by training data size is unfair for an RL method.

Authors' Response: The authors provided a new table with a detailed computational cost analysis, comparing RULE to baselines using FLOPS (floating-point operations), a much more direct and fair metric.

Final Decision Weight: This was highly impactful. The transparent and direct comparison using a standard metric fully resolved the reviewer's skepticism and converted a significant weakness into a well-supported claim. This directly led to Reviewer AitF raising their score.

Concern 3: Theoretical Soundness

Points Raised: Reviewer BvY9 identified an omission in an appendix proof, where the conditions for applying the martingale inequality had not been established.

Authors' Response: The authors acknowledged the omission and provided the missing assumptions along with a concise proof sketch, promising to include the full, rigorous proof in the final version.

Final Decision Weight: This demonstrated the authors' technical diligence. While a minor point, resolving it helped solidify the paper's standing as technically sound.

Concern 4: Inherent Limitations of the Approach

Points Raised: In a follow-up discussion, Reviewer esTn asked a crucial question: does the method truly erase information, or does it just teach a refusal behavior that could be bypassed by adversarial attacks?

Authors' Response: The authors ran new experiments against black-box and white-box attacks. They transparently reported that while the refusal behavior was robust, a strong gradient-based attack could still recover some information, acknowledging this as a limitation of the paradigm.

Final Decision Weight: This proactive and honest investigation was viewed very positively. Instead of overstating their claims, the authors characterized the boundaries of their method's effectiveness. This strengthened the paper's contribution by clearly positioning it within the field and highlighting areas for future work, which satisfied the reviewer.

Overall, the final decision was heavily influenced by the authors' thorough and scientifically rigorous rebuttal. They converted nearly every major weakness into a resolved point or a well-characterized limitation, which ultimately earned the reviewers' confidence and support for acceptance.

Reasons for recommendation:

The primary reason for recommending acceptance is the paper's novel and effective contribution to the important field of LLM unlearning. The work is technically solid, and the empirical results are strong.

While this is a strong paper, it does not meet the exceptional criteria required for a spotlight recommendation. None of the reviewers considered it as a groundbreaking contribution. Besides, even after the successful rebuttal, some valid reservations remained.